# A Light-weight Universal Medical Segmentation Network for Laptops Based on Knowledge Distillation

Songxiao Yang[1][0000−0001−9036−4817], Yizhou Li[1][0000−0002−7122−2087], Ye Chen[2][0009−0009−5564−1976], Zhuofeng Wu[1][0009−0005−4690−7403], and Masatoshi Okutomi[1][0000−0001−5787−0742]

[1] Tokyo Institute of Technology, Ookayama 2-12-1, Meguro, Tokyo, Japan
[2] The University of Tokyo, Kashiwanoha 5-1-5, Chiba, Japan
syang@ok.sc.e.titech.ac.jp, yli@ok.sc.e.titech.ac.jp,
chenye@g.ecc.u-tokyo.ac.jp, zwu@ok.sc.e.titech.ac.jp,
mxo@sc.e.titech.ac.jp

**Abstract.** In medical imaging, accurate and efficient segmentation is crucial for diagnostics, treatment planning, and monitoring disease progression. Traditional methods, while capable of providing reliable results, often require substantial computational resources, which may not be feasible on devices with limited capabilities such as standard CPUs and limited RAM. To address this challenge, we present an optimized universal segmentation framework that leverages a lightweight image encoder RepViT-M0.6, distilled from Swin-T. Our comprehensive analysis of the online validation set shows that our method surpasses the baseline LiteMedSAM model. We achieve a Dice Similarity Coefficient (DSC) of 84.68% and a Normalized Surface Dice (NSD) of 85.28%. Furthermore, the method achieves a more than threefold increase in inference speed, making it viable for real-time applications on devices with limited computational power. This demonstrates that our adaptation significantly enhances processing speed and resource efficiency without sacrificing accuracy.

**Keywords:** Segment Anything · Lightweight Model · Medical Imaging Segmentation · Computational Efficiency.

## 1 Introduction

Segmentation plays a crucial role in medical imaging analysis, involving the identification and delineation of regions of interest (ROI) within medical images. The precision of segmentation is crucial for numerous clinical tasks, including disease diagnosis, treatment planning, and monitoring disease progression [13, 50]. Traditionally, manual segmentation has been regarded as the standard for precisely defining anatomical and pathological regions. However, this method is highly time and labor-consuming and demands significant expertise. To overcome these

limitations, automatic segmentation techniques have been introduced. These advanced methods greatly reduce the required time and effort, improve consistency, and enable the efficient analysis of large-scale medical datasets [63].

Recently, deep-learning techniques for image segmentation have shown promising results by training networks to understand intricate image features and produce accurate segmentations [7]. However, many existing models designed for medical image segmentation face a significant limitation that they are tailored for specific tasks and may not perform well when applied to new tasks or different datasets [47]. This task-specific nature poses a challenge to the widespread use of these models in clinical settings. Conversely, recent advancements in natural image segmentation have introduced foundation models, like the segment anything (SAM) [34] and segment everything everywhere all at once [72], showing exceptional adaptability and performance across a range of segmentation tasks. Moreover, the development of MedSAM [42] aims to address the challenge of limited generalizability in medical image segmentation by facilitating universal segmentation across diverse medical imaging tasks.

Despite their strong performance, these methods often utilize large-scale image encoders, leading to high computational demands that limit their practicality. To address this issue and speed up inference while conserving resources, various approaches have been explored to replace the image encoder of SAM with lightweight models. For instance, MobileSAM [69] distills the knowledge of SAM's ViT-H model into a compact vision transformer, while EdgeSAM [71] employs a CNN-based model trained to mimic ViT-H, incorporating a meticulous distillation strategy with the prompt encoder and mask decoder. Additionally, EfficientSAM [66] leverages the MAE pretraining method to enhance performance. However, these methods typically suffer from significant performance drops.

In our work, we propose a solution to further accelerate inference and reduce resource usage while maintaining high performance. Firstly, we enhance the performance of the original LiteMedSAM by replacing its image encoder with Swin-T. Subsequently, to make the encoder lightweight, we distill a RepViT-M0.6 from Swin-T and substitute the encoder of Swin-T with the distilled RepViT-M0.6 image encoder, achieving higher speed and reduced resource consumption while preserving performance.

We extensively evaluate our proposal on the online validation set and compare it with the baseline model (LiteMedSAM). Our results demonstrate improved performance, with the evaluation metric DSC increasing by approximately 2% and NSD by around 1%. Furthermore, we achieve over three times faster inference speed on devices equipped with a CPU and 8GB of RAM.

## 2   Method

### 2.1   Pre-processing

We first conducted a statistical analysis of the challenge dataset. As shown in Table 1, Computed Tomography (CT) is the predominant modality, compris-

ing 76.70% of the dataset with 1,219,765 slices. Magnetic Resonance (MR) images are also well-represented, making up 13.55% with 214,454 slices. Positron Emission Tomography (PET) accounts for 4.03%, contributing 64,163 slices. Endoscopy and X-Ray images constitute smaller portions at 2.82% and 1.91% respectively, providing 44,804 and 30,360 slices. Other modalities such as Ultrasound(US), Dermoscopy, Microscopy, Optical Coherence Tomography (OCT), Mammography, and Fundus Photography each contribute less than 1%.

During the pre-processing of the external public datasets (the list of external public datasets is shown in Table 10), we initially excluded all slices or images lacking targets or containing extremely small targets (smaller than 20 pixels) to ensure each slice or image had at least one target for segmentation. Subsequently, we normalized all slices/images to a range of [0, 1] and stored each slice/image along with its corresponding ground truth in a single npy file to facilitate faster I/O operations.

In the training phase, all grayscale images were converted to 3-channel images by replicating the image three times along the channel dimension. We resized the longer side of all images to 256 pixels while maintaining the original aspect ratio and then padded them to $256 \times 256$ pixels to meet the input requirements of the encoder. If an image had multiple labels, one label was randomly selected. Random data augmentation was applied to both images and their corresponding ground truths. Additionally, we utilized multiple worker processes to accelerate data loading.

**Table 1.** Statistical analysis of the dataset.

| Modality | Proportion | Num. Slices |
|---|---|---|
| CT | 76.70% | 1219765 |
| MR | 13.55% | 215454 |
| PET | 4.03% | 64163 |
| Endoscopy | 2.82% | 44804 |
| X-Ray | 1.91% | 30360 |
| US | 0.40% | 6318 |
| Dermoscopy | 0.24% | 3874 |
| Microscopy | 0.11% | 1627 |
| OCT | 0.09% | 1436 |
| Mammography | 0.08% | 1233 |
| Fundus | 0.07% | 1100 |
| Total | | 1590134 |

### 2.2 Proposed Method

The proposed method employs a 2-stage training protocol for a teacher-student model. In the first stage, we train a strong teacher model by replacing MedSAM's image encoder with a Swin-T-based encoder. In the second stage, we distill the

features to a RepViT-M0.6-based MedSAM. The following sections will provide details of our model structures and the training & inference strategies.

**Teacher Model** As previous studies have shown [66,69,70], the image encoder is the heaviest and most parameter-intensive part of the SAM [34], significantly affecting segmentation performance. Thus, selecting a strong yet efficient image encoder is crucial. The default encoder for SAM is ViT-H [17], known for its strong capabilities. However, training SAM with ViT-H requires 68 hours on 256 A100 GPUs as mentioned in [34], posing a significant challenge for reproducibility or improvement.

To address this, we opt for the Swin-T image encoder [39], a small but effective hierarchical Transformer that uses shifted windows to limit self-attention computation to non-overlapping local windows while allowing cross-window connections. This architecture is efficient, modeling at various scales with linear computational complexity relative to image size, thus reducing the computational burden compared to ViT.

We replace lightweight MedSAM's original image encoder with Swin-T and train the entire pipeline from scratch. The Swin-T-based lightweight MedSAM shows significant improvement over the TinyViT-based lightweight MedSAM provided by the competition. Although Swin-T is the smallest Swin Transformer, it is still not efficient enough for fast inference on a laptop CPU. Therefore, we will use this model as a strong teacher model in the next section and distill its features into a smaller student model for much faster inference on a laptop CPU.

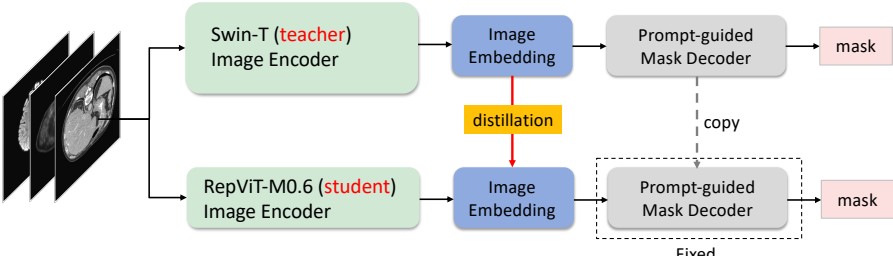

**Fig. 1.** Proposed teacher-student model architecture. For the teacher model (top), we use a Swin-T image encoder to replace the image encoder in the MedSAM and train the entire pipeline from scratch. For the student model (bottom), which is based on the RepViT-M0.6 image encoder, we distill the features from the teacher image encoder to the student image encoder. The prompt-guided mask decoder is directly copied from the teacher model and not finetuned.

**Student Model** With a well-trained teacher model, our next step is to select an efficient student model and effectively distill the teacher model's features into

it. For the student model, we choose RepViT-M0.6 [62], the smallest version of RepViT, as the image encoder for the student MedSAM model. RepViT is a series of lightweight CNNs redesigned from a ViT perspective, emphasizing their suitability for mobile devices. It builds on the mobile-friendly design of MobileNetV3 [28] and incorporates efficient architectural features of lightweight ViTs.

RepViT, being purely CNN-based, achieves very low latency and memory usage without the computational burden of attentions. After testing various RepViT variants, we found that RepViT-M0.6 offers sufficient performance for feature distillation with the highest inference speed on a CPU.

**Feature Distillation to Student Model** Next, we discuss the feature distillation from the teacher model to the student model. Following the practice in [69], SAM model distillation methods are classified into fully-coupled, semi-coupled, and decoupled distillation. The first two methods add supervision to the model's final output, i.e., the mask output, while decoupled distillation only distills the image encoder part.

Since the performance bottleneck mainly depends on the image encoder, it is reasonable to fix the prompt-guided mask decoder, which has a small number of parameters, and only distill the image encoder from the feature level. Therefore, we follow this practice and distill the image encoder part, as shown in Fig. 1, using a simple MSE loss between the outputs of the Swin-T encoder and the RepViT encoder. This simple distillation method works surprisingly well, with the student model's performance being comparable to the teacher model.

As mentioned in [69], finetuning the prompt-guided mask decoder after distilling the image encoder might potentially improve overall performance. However, in our case, the small image encoder with RepViT-M0.6 sufficiently matches the Swin-T in the feature level. Thus, finetuning the prompt-guided mask decoder with mask loss did not provide a performance boost.

**Loss functions.** For the teacher MedSAM model with Swin-T, we train the entire pipeline from scratch using a combination of Dice loss, cross-entropy loss, and MSE loss. This compound loss function is robust for various medical image segmentation tasks [41]. For the student MedSAM model with RepViT-M0.6, we distill only the image encoder part. We compute the MSE loss between the feature outputs of the teacher and student models' image encoders.

**Strategies to Accelerate CPU Inference** Our student model with RepViT-M0.6 is already fast on CPU inference. However, we explored quantization for potential benefits. We tried Pytorch FX Graph Quantization [6] and ONNX Runtime [15] for Int8 quantization, but observed significant performance loss even after calibration. Therefore, we abandoned quantization. Instead, we used "torch.jit" to increase the model loading speed, contributing to the speed boost in the Docker test.

### 2.3   Post-processing

First, as the model outputs are logits, we first convert the logits to probabilities using a sigmoid function. The masks are then cropped to match the shape of the image, which has been resized to have a longest side of 256 pixels, without being padded to $256 \times 256$ pixels. Following this, the masks are resized back to the original image dimensions using bilinear interpolation, ensuring proper alignment and smooth transitions. The resulting tensor is then converted to a NumPy array on the CPU. Finally, a threshold is applied to generate a binary segmentation mask, where values greater than 0.5 indicate the presence of the target object. This comprehensive process ensures that the model's raw outputs are accurately transformed into a practical segmentation format.

## 3   Experiments

### 3.1   Dataset and evaluation measures

We use the challenge dataset and external public datasets for network training, as shown in the supplementary Table 10.

The evaluation metrics include two accuracy measures—Dice Similarity Coefficient (DSC) and Normalized Surface Dice (NSD)—and the running time for efficiency measurement. These metrics collectively contribute to the ranking computation.

### 3.2   Environment settings

The details of our environment are presented in Table 2. We use Ubuntu 22.04.4 LTS as our operating system. Our system is equipped with an Intel(R) Core(TM) i9-13900KF CPU and 64GB of RAM. Additionally, we utilize an NVIDIA RTX 4090 GPU with 24GB of memory.

**Table 2.** Development environments and requirements.

| | |
|---|---|
| System | Ubuntu 22.04.4 LTS |
| CPU | Intel(R) Core(TM) i9-13900KF CPU@3.00GHz |
| RAM | 4×16GB; 2.67MT/s |
| GPU (number and type) | One NVIDIA RTX 4090 24G |
| CUDA version | 12.1 |
| Programming language | Python 3.10 |
| Deep learning framework | torch 2.1.2, torchvision 0.16.2 |
| Specific dependencies | N/A |
| Code | GitHub |

### 3.3  Training protocols of LiteMedSAM with Swin-T image encoder

In training LiteMedSAM with the Swin-T image encoder, we initially apply data augmentation techniques to enhance model robustness. These techniques include random horizontal and vertical flips. To avoid overfitting to the data sequence, we randomly select images from the dataset. For images with multiple labels, we randomly choose one label per image. The bounding box is generated by calculating the coordinates of the top-left and bottom-right corners of the label and applying a slight perturbation to them. A validation set is constructed by randomly selecting approximately 5% of the entire training dataset.

As shown in Table 3, during training, images are pre-processed to $3 \times 256 \times 256$. The network is trained from scratch over 100 epochs. We employ a combination of Dice Loss, Cross Entropy Loss, and Mean Squared Error Loss (MSELoss) as the loss function. The initial learning rate is set to 0.005. We use AdamW as the optimizer and ReduceLROnPlateau as the learning rate scheduler, which reduces the learning rate by a factor of 0.9 whenever the validation loss does not decrease for five consecutive epochs. We assess the model's performance on the validation set at the end of each epoch and save the model that records the best performance on the validation set.

**Table 3.** Training protocols of LiteMedSAM with Swin-T image encoder.

| | |
|---|---|
| Pre-trained Model | N/A |
| Batch size | 8 |
| Patch size | $3 \times 256 \times 256$ |
| Total epochs | 100 |
| Optimizer | AdamW |
| Initial learning rate (lr) | 0.005 |
| Lr decay schedule | ReduceLROnPlateau(reduction ratio 0.9) |
| Training time | 1200 hours |
| Loss function | Dice Loss, Cross Entropy Loss, MSE Loss |
| Number of model parameters | 14.55M |
| Number of flops | 42.85G |
| $CO_2$eq | 848 Kg |

### 3.4  Training protocols for the knowledge distillation of RepViT-M0.6 image encoder from Swin-T image encoder

For the knowledge distillation of the RepViT-M0.6 image encoder from the Swin-T image encoder, the training process is similar to the training of LiteMedSAM with Swin-T. The data is processed by data augmentation and shuffled before input into the network. We maintain the same dataset split as used in the training of LiteMedSAM with Swin-T. During training, images are pre-processed to $3 \times$

$256 \times 256$ and we conduct the distillation of the RepViT-M0.6 from the Swin-T image encoder over 50 epochs, as illustrated in Table 4. To minimize the difference in the image embedding outputs between RepViT-M0.6 and Swin-T, we calculate MSELoss. We utilize AdamW as the optimizer with an initial learning rate of 0.005 and ReduceLROnPlateau as the learning rate scheduler, which reduces the learning rate by a factor of 0.9 whenever the validation loss does not decrease for five epochs. We evaluate the model on the validation set after each epoch and save the model version that achieved the lowest validation loss.

**Table 4.** Training protocols for the knowledge distillation of RepViT-M0.6 image encoder from Swin-T image encoder.

| | |
|---|---|
| Pre-trained Teacher Model | Swin-T Image Encoder |
| Pre-trained Student Model | N/A |
| Batch size | 8 |
| Patch size | $3 \times 256 \times 256$ |
| Total epochs | 50 |
| Optimizer | AdamW |
| Initial learning rate (lr) | 0.005 |
| Lr decay schedule | ReduceLROnPlateau(reduction ratio 0.9) |
| Training time | 400 hours |
| Loss function | MSE Loss |
| Number of model parameters | 2.32M |
| Number of flops | 9.00G |
| $CO_2$eq | 99 Kg |

## 4   Results and discussion

### 4.1   Quantitative results on online validation set

In Table 5, we compare three methods: the baseline, LiteMedSAM with Swin-T, and our proposed LiteMedSAM with RepViT-M0.6 image encoder, which is distilled from the Swin-T model. We evaluate their performance on the online validation set using the DSC and NSD evaluation metrics.

The LiteMedSAM with Swin-T (without knowledge distillation) demonstrates an average improvement of approximately 2% in both DSC and NSD compared to the baseline. This improvement is observed across most modalities, with the exception of a slight decrease in Endoscopy and Fundus. Furthermore, when employing knowledge distillation, there is only a minor decline in the average DSC and NSD, yet still shows a clear improvement compared to the baseline.

**Table 5.** Quantitative evaluation results on online validation set.

| Target | Baseline | | w/o Knowledge Distillation | | Proposed | |
|---|---|---|---|---|---|---|
| | DSC(%) | NSD(%) | DSC(%) | NSD(%) | DSC(%) | NSD (%) |
| CT | 89.53 | 91.82 | 89.94 | 91.85 | **92.65** | **95.06** |
| MR | 78.75 | 81.87 | 81.35 | 84.36 | **86.09** | **89.34** |
| PET | 68.91 | 55.43 | **71.00** | **55.95** | 62.38 | 38.58 |
| US | 81.34 | 87.12 | 81.60 | 86.74 | **82.45** | **87.54** |
| X-Ray | 70.23 | 76.58 | 78.39 | **84.49** | **79.13** | 85.11 |
| Dermoscopy | 92.65 | 94.14 | **93.58** | **95.08** | 93.45 | 94.96 |
| Endoscopy | **94.87** | **97.38** | 93.87 | 96.43 | 93.48 | 96.23 |
| Fundus | **95.85** | **97.48** | 95.47 | 97.11 | 94.68 | 96.36 |
| Microscopy | 71.79 | 76.95 | 77.27 | 83.88 | **77.80** | **84.38** |
| Average | 82.66 | 84.31 | **84.72** | **86.21** | 84.68 | 85.28 |

**Table 6.** Quantitative evaluation of segmentation efficiency in terms of running time (s).

| Case ID | Size | Num. Objects | Baseline | w/o Knowledge Distillation | Proposed |
|---|---|---|---|---|---|
| 3DBox_CT_0566 | (287, 512, 512) | 6 | 206.4344 | 212.4705 | **48.5365** |
| 3DBox_CT_0888 | (237, 512, 512) | 6 | 55.7089 | 55.686 | **13.6685** |
| 3DBox_CT_0860 | (246, 512, 512) | 1 | 7.7789 | 7.6037 | **2.502** |
| 3DBox_MR_0621 | (115, 400, 400) | 6 | 101.7202 | 89.6444 | **20.3509** |
| 3DBox_MR_0121 | (64, 290, 320) | 6 | 58.291 | 50.8915 | **13.5169** |
| 3DBox_MR_0179 | (84, 512, 512) | 1 | 8.1488 | 7.0312 | **1.9713** |
| 3DBox_PET_0001 | (264, 200, 200) | 1 | 5.511 | 3.8106 | **1.3485** |
| 2DBox_US_0525 | (256, 256, 3) | 1 | 0.4136 | 0.4332 | **0.1382** |
| 2DBox_X-Ray_0053 | (320, 640, 3) | 34 | 1.3110 | 1.3009 | **1.271** |
| 2DBox_Dermoscopy_0003 | (3024, 4032, 3) | 1 | 0.7331 | 0.6091 | **0.4595** |
| 2DBox_Endoscopy_0086 | (480, 560, 3) | 1 | 0.4311 | 0.4164 | **0.1533** |
| 2DBox_Fundus_0003 | (2048, 2048, 3) | 1 | 0.4785 | 0.3656 | **0.1998** |
| 2DBox_Microscope_0008 | (1536, 2040, 3) | 19 | 0.9869 | 0.9626 | **0.6367** |
| 2DBox_Microscope_0016 | (1920, 2560, 3) | 241 | 8.7242 | 8.6334 | **8.1791** |

### 4.2   Qualitative results on online validation set

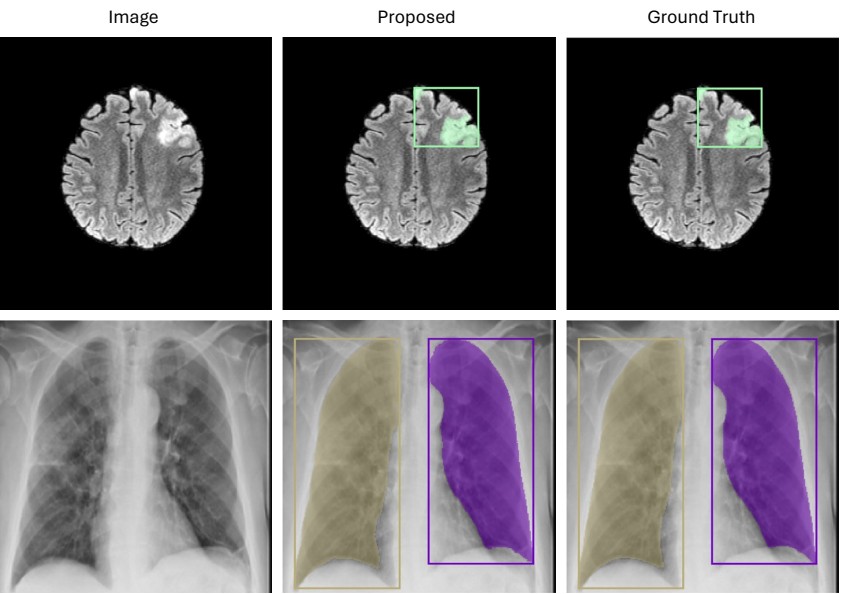

**Fig. 2.** The examples of good segmentation results.

In this section, we present examples of both good and bad segmentation results to further analyze the performance of our proposed method. Generally, our method performs well with images that exhibit high contrast, simple backgrounds, and regular shapes. However, it encounters challenges with images that contain overlapping textures and irregular shapes and borders.

As illustrated in Fig 2, in the first row, the brain MR image showcases high contrast against a simple background, while in the second row, the chest X-ray displays the lung in regular shapes, which our model can accurately segment. Conversely, as depicted in Fig 3, the X-ray image exhibits overlapping textures, and the targets within the box are irregularly shaped in both the first and second rows, posing difficulties for accurate segmentation with our method.

### 4.3   Segmentation efficiency results on online validation set

An important challenge for this task is the constraint of the target device, which is equipped with only a CPU and limited memory (8GB RAM), making segmentation efficiency crucial. We present some challenging cases that require longer processing times in Table 6. Our proposed method consistently demonstrates a significant reduction in running time across almost all cases compared to both the baseline and the LiteMedSAM with Swin-T. This improvement is particularly

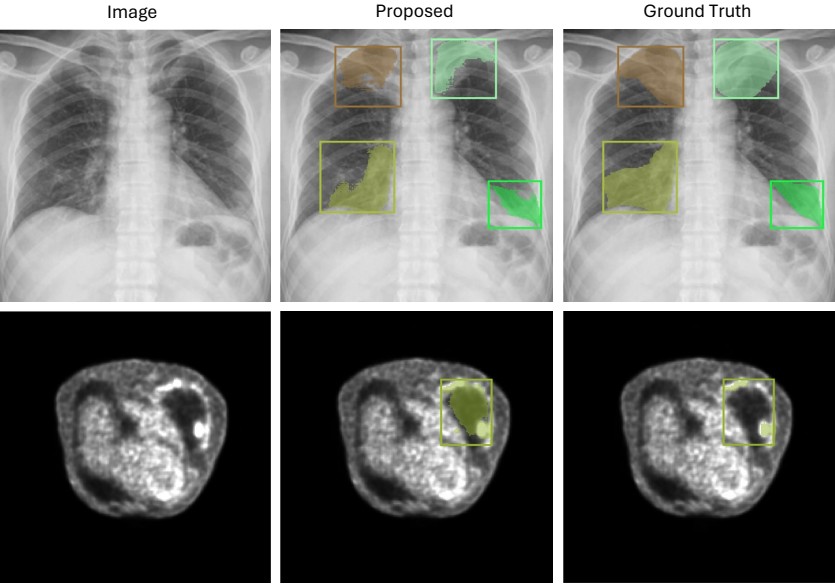

**Fig. 3.** The examples of bad segmentation results.

notable in complex 3D imaging cases. For instance, in the 3DBox_CT_0566 case, the proposed method reduces the running time from over 200 seconds in the baseline to just 48.54 seconds. Similarly, in the 3DBox_CT_0888 and 3DBox_CT_0860 cases, the running times decrease from 55.71 and 7.78 seconds in the baseline to 13.67 and 2.50 seconds, respectively.

In 2D imaging scenarios, such as 2DBox_US_0525 and 2DBox_X-Ray_0053, the proposed method drastically reduces running times to 0.14 and 1.27 seconds from 0.41 and 1.31 seconds in the baseline, respectively. The reductions are even more striking in cases like 2DBox_Dermoscopy_0003 and 2DBox_Endoscopy_0086, where the proposed method achieves running times of 0.46 and 0.15 seconds, down from 0.73 and 0.43 seconds in the baseline. These results underscore the effectiveness of our proposed method in enhancing the efficiency of segmentation on devices with limited computational resources.

### 4.4 Results on final testing set

The final testing set comprises 9 modalities: CT, MR, Endoscopy, Ultrasound(US), X-Ray, Fundus, Microscopy, PET, and OCT. Our proposed method is evaluated based on two categories of metrics: segmentation accuracy and segmentation efficiency. For accuracy, DSC and NSD are used, while segmentation efficiency is assessed by measuring the running time in seconds. A rank-then-aggregate strategy [65] is utilized for ranking. It includes the following three steps:

– Step 1: Compute the DSC, NSD, and running time for each test case.

– Step 2: Rank the teams for each modality based on each metric.
– Step 3: Calculate the average of all these rankings.

A quantitative comparison of our proposed method with the baseline LiteMed-SAM is presented in Table 7. Our method consistently outperforms LiteMedSAM in DSC and NSD across most imaging modalities. Specifically, our method significantly enhances NSD in US and Endoscopy, while markedly improving DSC in X-Ray and Endoscopy modalities. Moreover, our method achieves a reduction in running time for all modalities, thereby increasing both the efficiency and effectiveness of the segmentation.

Overall, our method achieves an average DSC of 79.26% and an average NSD of 81.16%, surpassing the baseline LiteMedSAM of 78.64% and 80.58%. The average running time is also reduced from 14.69 seconds with LiteMedSAM to 5.66 seconds, highlighting a substantial enhancement in processing speed.

Furthermore, our method scores an average ranking of 9.22, with a lower score indicating better performance, placing it 5th overall in the final test set.

**Table 7.** Quantitative evaluation results on final testing set.

| Target | LiteMedSAM(Baseline) | | | Proposed | | |
|---|---|---|---|---|---|---|
| | DSC(%) | NSD(%) | RunTime(s) | DSC(%) | NSD(%) | RunTime(s) |
| CT | 55.75 | 58.48 | 38.78 | **71.04** | **74.64** | **11.33** |
| MR | 64.80 | **62.75** | 18.57 | **67.95** | 62.10 | **6.45** |
| PET | 76.94 | 66.98 | 14.90 | **79.26** | **69.09** | **6.30** |
| US | 85.24 | 89.73 | 8.96 | **87.81** | **92.44** | **4.57** |
| X-Ray | **85.51** | **94.40** | 9.95 | 75.59 | 86.61 | **4.68** |
| Endoscopy | **94.41** | **96.95** | 7.56 | 92.32 | 95.05 | **4.12** |
| Fundus | 87.47 | 89.58 | 8.77 | **89.55** | **91.71** | **4.22** |
| Microscopy | **84.36** | **86.15** | 16.34 | 69.75 | 71.45 | **4.64** |
| OCT | 73.31 | 80.20 | 8.39 | **80.06** | **87.38** | **4.64** |
| Average | 78.64 | 80.58 | 14.69 | **79.26** | **81.16** | **5.66** |

### 4.5   Limitation and future work

Although our proposed method has exhibited promising performance and excellent efficiency, it still encounters difficulties with cases characterized by low contrast, irregular shapes, and overlapping textures, as mentioned in Section 4.2. These challenges highlight areas for improvement in future work.

One potential avenue for addressing these issues could involve training a more robust teacher model to provide better guidance during knowledge distillation. Additionally, incorporating more diverse and challenging data into the training process could help the model learn to handle such cases more effectively.

# 5 Conclusion

In this study, we introduced an optimized segmentation framework leveraging knowledge distillation techniques to enhance the efficiency of medical segmentation networks. By integrating knowledge distillation, notable reductions in inference time were achieved while preserving segmentation accuracy, thus demonstrating promising prospects for real-time application in resource-constrained environments. Specifically, our results indicate a significant threefold decrease in processing time compared to the baseline model, along with improvements in quantitative evaluation.

Despite these advancements, challenges remain with low-contrast images, irregular shapes, and overlapping textures. Future work will aim to address these issues, enhancing the robustness and applicability of our proposed framework.

**Acknowledgements** We thank all the data owners for making the medical images publicly available and CodaLab [67] for hosting the challenge platform.

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

**Table 8.** Checklist Table. Please fill out this checklist table in the answer column.

| Requirements | Answer |
|---|---|
| A meaningful title | Yes |
| The number of authors ($\leq 6$) | 5 |
| Author affiliations and ORCID | Yes |
| Corresponding author email is presented | Yes |
| Validation scores are presented in the abstract | Yes |
| Introduction includes at least three parts: background, related work, and motivation | Yes |
| A pipeline/network figure is provided | Figure 1 |
| Pre-processing | Page 2 |
| Strategies to data augmentation | Page 7 |
| Strategies to improve model inference | Page 5 |
| Post-processing | Page 6 |
| Environment setting table is provided | Table 2 |
| Training protocol table is provided | Table 3 4 |
| Ablation study | Page 9 |
| Efficiency evaluation results are provided | Table 6 |
| Visualized segmentation example is provided | Figure 2 3 |
| Limitation and future work are presented | Yes |
| Reference format is consistent. | Yes |
| Main text $>= 8$ pages (not include references and appendix) | Yes |

**Table 9.** The challenge datasets we used for training.

| Dataset | Anatomy | Modality |
|---|---|---|
| COVID-19-20 [56] | Chest | CT |
| AbdomenCT-1K [45] | Abdomen | CT |
| FDG-PET-CT-Lesions [20] | Whole body | CT |
| NSCLC Radiogenomics [8] | Chest | CT |
| NSCLC-Radiomics [1] | Lung | CT |
| CT Lymph Nodes [55] | Abdomen, Mediastinum | CT |
| NSCLC-PleuralEffusion [35] | Chest | CT |
| NSCLC-LungMSD-LUNG [7] | Chest | CT |
| KiTS23 [24] | Kidney | CT |
| CT-ORG [54] | whole body | CT |
| COVID-19-20-CTSEG [43] | Chest | CT |
| TotalSegmentator [64] | whole body | CT |
| AMOS [31] | Abdomen | CT, MR |
| LCTSC [68] | Chest | CT |
| HCC-TACE-Seg [48] | Liver | CT |
| Adrenal-ACC-Ki67-Seg [2] | Abdomen | CT |
| MSD [7] | various | CT, MR |
| ISLES [25] | Brain | MR |
| WMH [37] | Brain | MR |
| BraTS [46] | Head | MR |
| PROMISE12 [38] | Prostate | MR |
| MSD-Prostate [57] | Prostate | MR |
| NCI-ISBI [12] | Prostate | MR |
| Crossmoda [16] | Brain | MR |
| QIN-PROSTATE-Repeatability [18] | Prostate | MRI |
| CC-Tumor Heterogeneity [9] | Cervical Cancer | MR |
| COVID-19 Radiography Database [53] | Lung | CXR |
| COVID-QU-Ex [60] | Lung | CXR |
| Chest Xray Masks and Labels [29] | Chest | CXR |
| Chest X-Ray Images with Pneumothorax Masks | Chest | CXR |
| CDD-CESM [33] | Breast | Mammography |
| Intraretinal Cystoid Fluid [3] | Eye | OCT |
| ps-fh-aop-2023 [40] | Head | US |
| hc18 [26] | Fetal Head | US |
| Breast Ultrasound Images Dataset [4] | Breast | US |
| ISIC2018 [61] | Skin | Dermoscopy |
| CholecSeg8k [27] | Abdominal | Endoscopy |
| Kvasir-SEG [30] | Abdominal | Endoscopy |
| m2caiSeg | Abdominal | Endoscopy |
| PAPILA [36] | Eye | Fundus |
| IDRiD [52] | Eye | Fundus |
| NeurIPS CellSeg [44] | Cells | Microscopy |

**Table 10.** The external public datasets we used for training.

| Dataset | Anatomy | Modality |
|---|---|---|
| CT Lung & Hearth & Trachea segmentation | Chest | CT |
| Seg.A. [49] | Aorta | CT |
| Figshare Brain Tumor Dataset [11] | Brain | MR |
| Uwaterloo skin cancer [21] | Skin | Dermoscopy |
| BKAI-IGH NeoPolyp [5] | Abdominal | Endoscopy |
| CT2USforKidneySeg [59] | Kidney | US |
| Ultrasound Nerve Segmentation | Neck | US |
| GlaS@MICCAI'2015: Gland Segmentation [58] | ColonGland | Microscopy |
| MM-WHS [19] | Heart | CT, MR |
| MMs-20-21 [10] | Heart | MR |
| FeTA [51] | Brain | MR |
| CHAOS [32] | Abdomen | CT, MR |
| Drive | Eye | Fundus |
| RAVIR [22, 23] | Eye | Fundus |
| FetoPlac | Fetus | Microscopy |
| HMC-QU [14] | Heart | US |