# OpenReview forum: "A Light-weight Universal Medical Segmentation Network for Laptops Based on Knowledge Distillation"
_thecvf.com/CVPR/2024/Workshop/MedSAMonLaptop — CVPR24 MedSAMonLaptop_

### Official Review · Reviewer_Qnph · 2024-06-12
**Review of "A Light-weight Universal Medical Segmentation Network for Laptops Based on Knowledge Distillation"**

**Rating:** 8
**Confidence:** 5

**Review:**

Summary:

The paper describes the development of a segmentation network, designed in the light of the “CVPR 2024: Segment Anything In Medical Images On Laptop” Challenge. In contrast to the original SAM and MedSAM publications, the authors train a SAM-like model with a SWIN-T backbone from scratch on the provided and external datasets. Further distilling the SWIN-T encoder to a lightweight RepViT-M0.6 encoder results in a performant and efficient model, beating the LiteMedSAM baseline in all three metrics - DSC, NSD and inference speed.

Strengths:
- Simplicity: The contribution leverages well-known encoder backbones and concepts for knowledge distillation, resulting in an efficient yet high-performing final model.
- Resource Efficiency: The non-distilled network already matches the inference speed of the LiteMedSAM baseline. Moreover, the proposed model is not only efficient during inference but also significantly more efficient during training compared to the original SAM architecture.
- Reproducibility: The code for the challenge submission is available on GitHub, ensuring complete reproducibility.

Weaknesses:
- Comparison to MedSAM: The network is trained on a larger dataset with additional data sources, as detailed in Table 9. It is unclear whether the improvements of the proposed model are due to this larger training dataset or the different architecture and training pipeline. This question is particularly interesting since the proposed model does not use the pretrained SAM checkpoint but is trained from scratch.
- Size of External Dataset: It would be beneficial to include the dataset size of the external data, similar to the information provided in Table 1.

Overall:
The submission represents a solid contribution to the challenge, improving upon the LiteMedSAM baseline in both performance and efficiency. The mentioned weaknesses are either outside the scope of a challenge submission or could be easily addressed.

---

### Official Review · Reviewer_gWv1 · 2024-06-15
**Knowledge Distillation approach for medical image segmentation using foundation models**

**Rating:** 8
**Confidence:** 5

**Review:**

## Summary

In **"Light-weight Universal Medical Segmentation Network for Laptops Based on Knowledge Distillation,"** the authors propose a model distillation approach for optimising the applicability of foundation models for medical image segmentation on edge devices. Specifically, their approach consists of two steps. The first consists of replacing the image encoder from LiteMedSAM with Swim-T. Then, the entire pipeline (I assume it means the new image encoder in addition to the prompt encoder and mask decoder, but could you mention that explicitly?) is trained from scratch. In the second step, to optimise inference runtime, they distil the teacher (Swim-T) model's feature into a student model (RepViT-M0.6), a CNN-based model that allows for faster computation. They report that their approach is 3-fold faster than the challenge-provided baseline, while the performance is equivalent (if not improved). Overall, this is a very well-written and clear manuscript, and I have only a few minor comments/suggestions to improve the manuscript's **completeness** and **reproducibility**.
## Detailed Comments
**Abstract:**
1 – Could you report the DSC and NSD scores explicitly instead of relative to LiteMedSAM? Without knowing the performance of LiteMedSAM, "improvements by 2%" are meaningless.
**Methods:**
Overall, the methods are clear. However, there are some inconsistencies or missing information:
2 – Pre-processing: The authors share some descriptive statistics of the training dataset from the challenge (and, hence, from MedSAM work), but they don't say that explicitly. Someone who is not part of the challenge might be unaware of what the author means by "... statistical analysis on our dataset." So, I assume that all the data used is solely from the training data provided by the challenge? That is unclear as the authors used the term "external public datasets" after "our dataset." I'm assuming Table 1 shows all the training data used for model training (minus 5% for validation). The authors also forgot to reference Table 9 in the text.
3 – Proposed method-Teacher Model: There is a minor language error in "... significant challenge for **reproduction**...". I'd assume the author meant "reproducibility."
4 – Proposed method-Teacher Model: The authors say, "We replace MedSAM's original encoder..." and continue to mention MedSAM in that paragraph, but in the rest of the manuscript, they report that LiteMedSAM's image encoder is replaced. Which one is the case?
5 – Proposed method-Loss function: Here, there are other inconsistencies about the original model used (MedSAM or LiteMedSAM) and the loss function used for training the teacher model, where it is said "... we train the entire pipeline from scratch using a combination of Dice loss and Focal loss", but in table 3 it says: "Dice Loss, Cross Entropy Loss, MSE Loss." Which one is it? Could you perhaps provide a formula or links to PyTorch functions used?
6 – Proposed method-Post-processing: You mention that after the logits are transformed to probabilities, "the masks are then cropped to match the new size". I would assume the output has the same 256x256 size as the resized and padded input image. Could you clarify that?
**Experiments:**
7 – Training protocols of LiteMedSAM with Swin-T image encoder: A key missing information is about how the bounding box was generated for model training.
**Results and Discussion:**
8 – Quantitative results on online validation set: In "The LiteMedSAM (without knowledge distillation) ..." I assume the authors are referring to LiteMedSAM with Swin-T. Could you rephrase that?

Apart from these suggestions for improvement, the authors did a nice job in their proposed approach and manuscript. The authors met all the requirements from the checklist, but in the checklist table, the references to tables with environment settings and training protocols need to be corrected.

As a final note, the authors’ GitHub repository looks well organised, but I haven’t tried to run the code myself.

---

### Official Review · Reviewer_nHuE · 2024-06-15
**Well-structured article with minor errors**

**Rating:** 7
**Confidence:** 4

**Review:**

Pros:
- The paper is well-structured and provides a comprehensive overview of the development and evaluation of the approach. The methodology is also clearly presented.

Cons:
- In the introduction, you have mentioned MobileSAM, EdgeSAM, EfficientSAM, and EfficientViT-SAM as examples of lightweight models and stated that these models suffer from performance drops. This is not true for EfficientViT-SAM. Its benchmarks show that it has competitive results compared to SAM-ViT-H, which is the largest variant of SAM.
- In the data preprocessing section, there are 11 modalities in total, it seems like you have missed the microscopy type.

---

### Decision · Program_Chairs · 2024-10-01

Accept